# Relationship between Unilateral Posterior Crossbite and Human Static Body Posture

**DOI:** 10.3390/ijerph17155303

**Published:** 2020-07-23

**Authors:** Jorge Zurita-Hernandez, Raul Ayuso-Montero, Meritxell Cuartero-Balana, Eva Willaert, Jordi Martinez-Gomis

**Affiliations:** 1Department of Prosthodontics, School of Dentistry, Faculty of Medicine and Health Sciences, University of Barcelona, 08907 Catalonia, Spain; jzurithe@live.com (J.Z.-H.); meritxellcuartero@gmail.com (M.C.-B.); evawillaert@ub.edu (E.W.); jmartinezgomis@ub.edu (J.M.-G.); 2Oral Health and Masticatory System Group (Bellvitge Biomedical Research Institute) IDIBELL, L’Hospitalet de Llobregat, Barcelona, 08907 Catalonia, Spain

**Keywords:** dental occlusion, body posture, malocclusion, crossbite, photogrammetry, mandibular position

## Abstract

Background: We compared photogrammetry-assessed body posture between young adults with and without unilateral posterior crossbite (UPCB). Assessments were controlled by vision, mandibular position and sitting/standing position. In addition, we aimed to determine the relationship between UPCB laterality and the direction of body posture using photogrammetry and a static postural platform. Methods: Adults with natural dentition, with and without UPCB, were enrolled. Static body posture was assessed by photogrammetry based on horizontal acromial alignment and horizontal anterior-superior iliac spine (ASIS) alignment. Frontal photographs were taken with participants asked to open or close their eyes and hold their jaws at rest, at an intercuspal position, and at left or right lateral positions. Distribution of foot pressure was recorded using a static postural platform at different visual input and mandibular positions. General linear models with repeated measures were used to assess the effect of the various within- and between-subject factors. Results: In total, 36 adults (left UPCB = 12; Right UPCB = 6; controls = 18) participated. There were significant differences between the control and UPCB groups in horizontal alignment at the acromion (*p* = 0.035) and ASIS (*p* = 0.026) levels when controlled by visual input and mandibular position. No significant differences in horizontal alignment or foot pressure distribution were observed by laterality in the UPCB group. Conclusion: The presence of UPCB affects static body posture, but the side of crossbite is not related to the direction of effect on static body posture.

## 1. Introduction

Unilateral posterior crossbite (UPCB) at the intercuspal position (ICP) refers to an occlusal relationship in which the mandibular teeth of only one side are located facial to the opposing maxillary teeth. The condition is thought to affect as much as 12% of the general population [1]. Several studies have related the presence of this type of asymmetry to occlusal interference, abnormal mandible growth and asymmetric muscular activity [2,3,4]. However, other studies have reported that UPCB does not contribute to asymmetric jaw muscle activity during functional tasks [5] nor to a preferred chewing side [6,7]. It has also been reported that UPCB could be associated with the body posture, especially with postural control and with scoliosis [8,9], though no association has been shown between the side of the scoliosis and the side of the posterior crossbite [8]. In other cross-sectional studies, midline-displacement or crossbite were found to have no influence on postural control or plantar pressure distribution [10,11,12]. These apparent discrepancies could be explained by the different populations studied and by the methods used.

Body posture concerns the spatial relation between different anatomical parts of the human body and requires static and dynamic balance based on motor requirements and environment influences [13]. The static aspect of body posture is usually assessed with static postural platforms, but the specificity and sensitivity of these instruments are very poor [13,14,15]. Photographic measurement or photogrammetry is a more reliable tool for assessing cervical posture in both the standing and sitting positions [16,17]. Horizontal acromial alignment and anterior–superior iliac spine (ASIS) tool can be measured by this approach as a means of assessing trunk alignment in static body posture [18]. The sensory information received by the vestibular, visual and proprioceptive systems modulates postural balance, with deprivation of visual input potentially affecting body posture [13]. It has been reported that temporarily manipulated occlusion, different mandibular positions or sitting/standing positions might result in different body postures [12,19,20,21]. Therefore, when assessing body posture, different methods should be combined (e.g., postural platform and photogrammetry) and different sources of sensory input should be controlled (e.g., visual input, mandibular position and sitting/standing position [14,22,23]. If body postures are shown to differ between people with and without UPCB, and if the side affected by UPCB correlates to the direction of body posture alteration, this may be necessary to indicate a possible causative association.

The primary aim of this study was to compare body posture assessed by photogrammetry in young adults with and without UPCB, controlling for vision, mandibular position and resting (sitting/standing) position. A secondary aim was to determine the relationship between UPCB laterality and the direction of body posture alteration, using both photogrammetry and a static postural platform. Finally, as a third aim, associations were explored between the side affected by UPCB and several functional lateralities.

## 2. Materials and Methods

### 2.1. Participants

We enrolled healthy adults with and without UPCB from a cohort of volunteer students at the University of Barcelona Dental School (Catalonia, Spain) and from friends or relatives of students and staff of the school. The inclusion criteria were young adults with a minimum of 28 natural teeth, and for the unilateral crossbite group (UPCB), they had at least 1 posterior tooth in crossbite on only one side. The exclusion criteria were auditory, visual, vestibular, neurological, temporomandibular or painful musculoskeletal disorders; pharmacological treatments and history of orthodontic treatment. All participants signed an informed consent form that had been approved by the local Ethics Committee (Code 2015/21). The study was carried out in accordance with the principles of the Helsinki Declaration.

### 2.2. Intervention and Data Collection

Data for gender, age, height and weight were obtained by an interview with each participant. The side and the number of posterior teeth in cross bite, and the direction and amount of midline deviation, were recorded from a clinical examination. A single researcher performed all tests of laterality, as follows: hand (observing that used to stir liquid in a glass), foot (observing that used to stamp on paper placed on the floor), ear (observing that used to listen through a hole) and eye (observing that used to look into a dark bottle) [24]. Finally, masticatory laterality was quantitatively assessed by asymmetry index (AI), using a 10 cm VAS (visual analogue scale) between the left end equivalent to ‘chewing always left’ (−1), and the right end as ‘chewing always right’ (+1), with ‘no preference’ (0) in the middle [25,26]. The reliability of these tests was reported high previously [25].

Static body posture was measured according to the protocol of Ferreira et al. [18], using a digital camera (Canon, model 1000D, Tokyo, Japan), a 63-cm tripod and 15-mm polystyrene balls in an acoustic-free controlled environment. The camera was placed on the tripod 3 m from the subject, with ground marks in place to centre the tripod to the static postural platform (Slim sensor one Milletrix v. 1.0.0.26; Diasu; Roma, Italy) when taking photographs in the standing posture. Other ground marks were placed to centre the tripod to the stretcher when taking photographs while the participant was seated. Additionally, two markers separated by 80 cm were placed in a vertical plumb to calibrate the images. Seven anatomical points were marked using polystyrene balls fixed with a double-sided adhesive tape: one on the spinous process of seventh cervical vertebrae (C7) and three each bilaterally on the tragus, acromion and ASIS (Figure 1).

All participants were taught to move the jaw in four different positions (rest, ICP, left-lateral and right lateral). To standardize each lateral position, three marks were made on the maxillary central incisors, with the central mark indicating the midline of the mandibular arch and the other two marks placed 3 mm to the left and right using a template [27]. Photographs were taken in frontal and right- and left-lateral directions, asking participants to perform various activities: opening or closing their eyes to control for vision; holding their jaws in rest, ICP and left- and right- lateral positions to control for mandibular position and standing in the static postural platform or seated in the stretcher in order to control for the sitting/standing position. To minimize risk of bias each photograph was taken after 50 s to stabilize each position. Distribution of foot pressure was recorded just after taking frontal photographs of the participants standing on the static postural platform at the different visual and mandibular positions.

### 2.3. Data Analysis

Photographs were analyzed using open-source postural analysis software (PAS/SAPO, Universidade de Sao Paolo, Sao Paolo, Brazil) [28]. A line connecting each bilateral marker (acromial or ASIS) was drawn and angles were calculated with reference to the horizontal line. Positive values were in the anticlockwise direction and negative values were in the clockwise direction (Figure 1a). The angle between the reference horizontal line (90° from the vertical plumb), and the line formed by the tragus and C7 was also measured (Figure 1b). Distribution of foot pressure recorded from the static postural platform was expressed as a percentage of weight distribution to the left foot. The same operator conducted all procedures (JZ).

Sample size was determined based on the primary aim considering a type I error of 0.05, a power of 0.8, a standard deviation of 2.0° based on prior research [18] and an estimated dropout of 10%, seeking to find a clinical difference of 2° in acromion alignment. Therefore, 18 subjects were needed in each study group (crossbite and control groups). A test–retest assessment was performed in 12 participants based on their availability. Intraclass correlation coefficients for the main parameters were determined using a mixed model with a random effect for the individual. To calculate the measurement error, the Dahlberg’s formula was applied [29]. Kruskal–Wallis tests, adjusted for multiple tests by Bonferroni correction or the Chi-square test were used to analyze differences in demographic characteristics and functional lateralities between control, left UPCB and right UPCB groups. Four general linear models with repeated measures were used to assess acromion and ASIS horizontal alignment, the cervical/tragus (CT) angle and the weight-foot distribution as dependent variables. Visual input, mandibular position and when appropriate, standing position and the projection were considered within-subject factors. As a between-subject factors was first considered as control vs. UPCB groups and then left UPCB, right UPCB and control groups. For these, pairwise comparisons were performed based on the estimated marginal means and adjustment for multiple comparisons by a Bonferroni test. The significance level was set at *p* < 0.05 for all analyses.

## 3. Results

We enrolled 36 healthy adults, with 18 in the UPCB group (14 women and 4 men) and 18 in the control group (13 women and 5 men). All participants had a minimum of 28 natural teeth. Their details are summarized in Table 1. We were able to complete a test–retest assessment with 12 participants (11 women and 1 man, mean age 20 years), of whom 10 were from the control group and 2 were from the UPCB group). The intraclass correlation coefficients for these are shown in Table 2. The Dahlberg error in measuring the horizontal alignment of the acromions ranged, depending on the mandibular position, from 1.2 to 1.6° by standing posture and from 0.6 to 1.1° by seated posture, and from 1.0 to 1.3° measuring the horizontal alignment of the ASIS.

In the UPCB group, 12 participants had the crossbite on the left (9 had midline deviation to the left) and 6 had the crossbite on the right (5 had midline deviation to the right). Most participants showed a preference for their right sides in handedness, footedness, earedness and eyedness, with no significant differences between the control group and the right or left UPCB groups. Participants in the left UPCB group reported a different AI compared with the right UPCB group (*p* = 0.04; Kruskal–Wallis test).

The horizontal alignment data measured in a frontal view is shown in Figure 2 and Figure 3. Horizontal acromion alignment was significantly different between the UPCB and control groups (*p* = 0.010) when controlled by visual input (*p* = 0.045), mandibular position (*p* = 0.211) and standing/sitting position (*p* = 0.255). Horizontal acromion alignment was significantly different between the left UPCB, right UPCB and control groups (*p* = 0.035) when controlled by visual input (*p* = 0.123), mandibular position (*p* = 0.076) and standing/sitting position (*p* = 0.938; Figure 2). Mandibular position had no effect based on the side of the crossbite (interaction effect *p* = 0.225). Pairwise comparisons reveal that the control group had a no-significant different acromial angles than right UPCB group (mean difference, 1.66°; 95%CI, −0.26–3.59°; *p* = 0.110) and the left UPCB group (mean difference, 1.35°; 95%CI, −0.17–2.88°; *p* = 0.095). No significant difference was observed for horizontal acromial alignment between the right and left UPCB groups.

Horizontal ASIS alignment was significantly different between the UPCB and control groups (*p* = 0.008) when controlled by visual input (*p* = 0.151) and mandibular position (*p* = 0.577). Horizontal ASIS alignment was also significantly different between the left UPCB, right UPCB and control groups (*p* = 0.026). There was also no effect on horizontal alignment controlling for visual input (*p* = 0.199) and mandibular position (*p* = 0.908; Figure 3). In addition, mandibular position did not have different effects depending on the laterality of the crossbite (interaction effect *p* = 0.522). Pairwise comparisons revealed that control group had similar ASIS angles than the right UPCB group (mean difference, 2.02°; 95%CI, −0.06–4.09°; *p* = 0.060) and the left UPCB group (mean difference, 1.42°; 95%CI, −0.23–3.06°; *p* = 0.011). No significant difference was observed in the horizontal alignment of the ASIS between the right and left UPCB groups.

The side of the crossbite had no significant effect on the CT angle (*p* = 0.79), even after controlling for visual input (*p* = 0.007), mandibular position (*p* = 0.001), standing/sitting position (*p* < 0.0005) and projection (*p* = 0.175). The mean estimated values for all participants in each mandibular position were as follows: resting, 48.0° (95%CI, 46.6–49.4°); ICP, 47.6° (95%CI 46.2–48.9°); left-lateral, 49.0° (95%CI 47.7–50.4°) and right lateral, 49.0° (95%CI 47.6–50.4°). However, the difference was 4.1° (95%CI 3.3–4.9°) higher when standing compared to sitting and 0.5° (95%CI 0.1°; 0.8°) higher with open eyes than with closed eyes.

Finally, the side of crossbite was unrelated to weight distribution (*p* = 0.830) measured in the static postural platform. This remained when controlled for visual input (*p* = 0.492) and mandibular position (*p* = 0.338). The means for the control, left UPCB and right UPCB groups were 48.13% (95%CI, 46.9%–49.4%), 48.55% (95%CI, 47.0%–50.1%) and 47.81% (95%CI, 45.7%–49.9%), respectively.

## 4. Discussion

Our results indicate that individuals with and without UPCB have differences in static body postures when viewed from the front. These differences amounted to 1–2° in the horizontal alignment at the acromion and ASIS levels. Participants in the control group showed no balanced horizontal alignment of the acromions, achieving similar values to those reported elsewhere [18]. When assessing body posture by frontal photogrammetry, we noted that reliability ranged from moderate to excellent and similar to that reported for sagittal projection in other studies [16,17]. Moreover, the observed differences in body posture were detected after controlling for visual input, mandibular position and standing/sitting position. Therefore, our results support the research finding that individuals with UPCB might have different body postures to individuals without UPCB [8].

Although the difference in static body posture between the UPCB and control groups was statistically significant, the magnitude of difference (1–2°) was relatively small and could be of only marginal clinical relevance. Furthermore, the range of measurement errors was close to the magnitude of this difference. Indeed, the difference may be attributable to physiological muscular asymmetry compatible with normal function [2,15,30,31]. Participants showed similar static body postures irrespective of whether their crossbite was on the left or right, similar to the observation that the degree of vertebral deviation correlated with the degree of mandibular deviation only when excluding direction [32]. Although the sample size was sufficient to find body posture differences in subjects with and without UPCB, the small number of participants in the right-side or left-side UPCB group might not have been large enough to find significant comparisons and the findings should be interpreted with caution. Correcting or accentuating the mandibular position to the crossbite side also had no effect on static body posture, indicating that momentarily eliminating the possible cause did not eliminate the effect. In fact, no clinically relevant changes on body posture have been observed after correcting for the UPCB in two longitudinal studies [33,34]. Finally, not all studies with different designs have showed similar results for the magnitude and direction of static body posture based on different malocclusions or mandibular positions [11,12,15,20,33]. Many of the factors that should be taken into account to support a cause–effect relationship between UPCB and body posture were not observed in the present study, which brings our findings in agreement with those of a review study [35].

It is possible that individuals with UPCB have a different upper body posture to controls, but that they maintain a similar weight distribution through their feet, as registered on the platform. The present results contrast with the theory of dental occlusion as a potential factor influencing postural adaptations and support the position that there is a lack of solid evidence for a correlation between dental and orthopedic findings [15,36]. Consequently, as reported elsewhere, there remains no scientific evidence that correcting UPCB can prevent or treat postural disorders or musculoskeletal problems [10,15,35]. A well-designed prospective study with appropriate controls is warranted to clarify the effect of crossbite correction on different aspects of body posture.

Interestingly, subjects with UPCB reported that they tended to chew more on the side of their crossbite. A similar tendency, though not significant, was reported in another study in which the chewing side preference was assessed qualitatively [5]. Other cross-sectional studies have found no relationships between the crossbite side and the preferred chewing side [6,7]. However, subjects with crossbite have showed alterations in both contact glide distances and masticatory cycle morphology [37,38]. Masticatory pattern and masticatory laterality are important aspects of masticatory function that are influenced by multiple central or peripheral factors [24,25,26,39,40]. UPCB may be one such factor, but to date, treatment for UPCB has not been shown to alter the masticatory cycle shape [37] or preferred chewing side [5]. Furthermore, the present study failed to find any relationship between UPCB laterality and other functional lateralities. Further studies with appropriate sample size would be required to clarify the association between UPCB laterality, masticatory asymmetry and other functional lateralities.

The present study has several limitations. For example, only the immediate effect of modifying mandibular position was evaluated, and we could not simulate the effects of permanent changes in mandibular position after correcting the UPCB. Moreover, masticatory laterality was only assessed subjectively by the participant.

## 5. Conclusions

The presence of UPCB is associated with different static body postures. However, not only may these differences be clinically irrelevant but also the side of crossbite shows no relationship to the side of deviation in static body posture. We also showed that UPCB might be associated with a tendency to chew more on the side of the crossbite, although the affected side was not associated with other functional lateralities.

## Figures and Tables

**Figure 1 ijerph-17-05303-f001:**
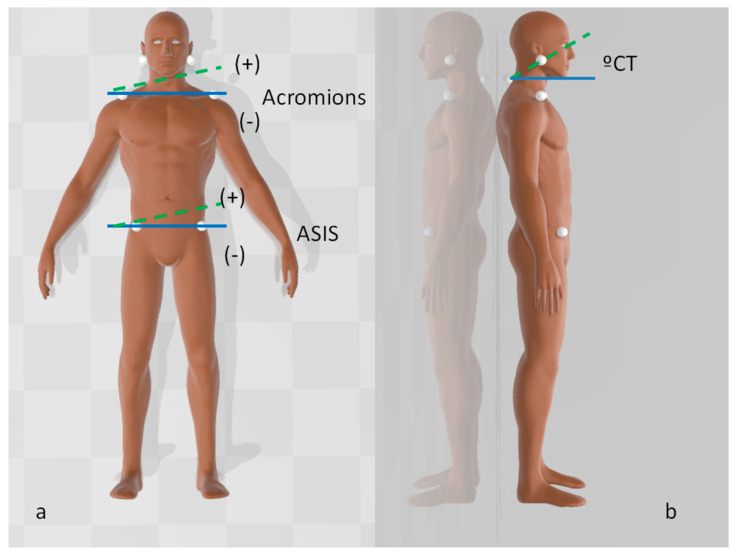
Anatomical points marked with polystyrene balls. (**a**) Horizontal alignment at each acromion and anterior-superior iliac spine (ASIS), as measured from the frontal view. (**b**) The cervical/tragus angle (°CT), as measured in the sagittal plane.

**Figure 2 ijerph-17-05303-f002:**
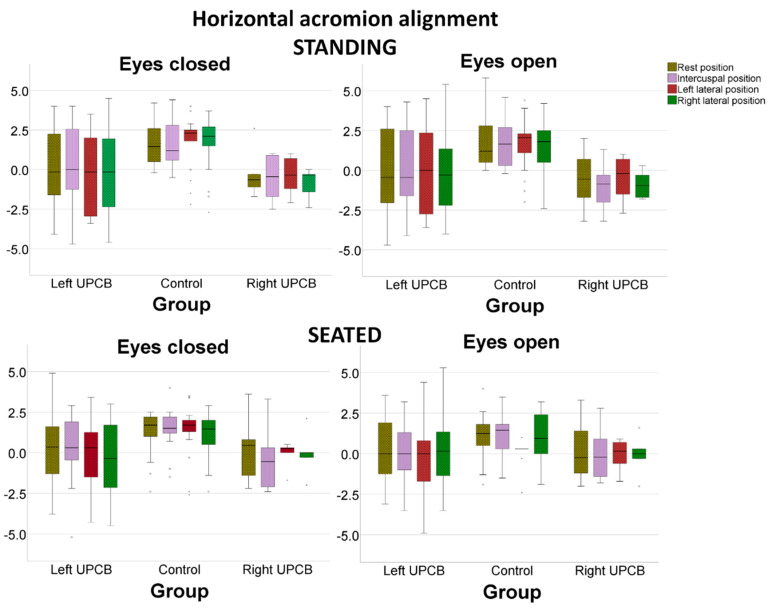
Horizontal acromion alignment (°), as controlled by visual input, by mandibular position and by standing/seated posture.

**Figure 3 ijerph-17-05303-f003:**
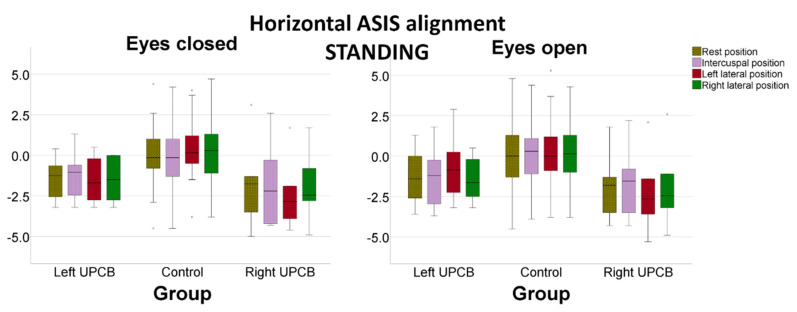
Horizontal anterior–superior iliac spine alignment (°), as controlled by visual input and by mandibular position.

**Table 1 ijerph-17-05303-t001:** Demographic characteristics and functional laterality of participants.

Data	Control (*n* = 18)	UPCB (*n* = 18)			*p*-Value
		Total (*n* = 18)	Left (*n* = 12)	Right (*n* = 6)	
Gender (% women)	77.8	72.2	75	66.7	0.86
Age (years)	20.8 (1.4)	23.4 (5.8)	25.0 (6.0)	20.3 (4.0)	0.09 †
Height (cm)	168.2 (10.2)	169.7 (10.5)	170.6 (11.8)	168.0 (8.2)	0.73 †
Weight (kg)	62.1 (10.3)	64.1 (10.6)	64.6 (10.6)	63.0 (11.6)	0.78 †
Crossed teeth (n)	0	3.0 (1.5)	3.3 (1.5)	2.3 (1.4)	
Midline deviation (mm)	0	2.94 (2.0)	2.75 (2.1)	3.33 (1.9)	
Handedness (% right)	88.9	88.9	83.3	100	0.32
Footedness (% right)	94.4	77.8	83.3	66.7	0.22
Earedness (% right)	88.9	61.1	66.7	50	0.12
Eyedness (% right)	61.1	55.6	58.3	50	0.89
Masticatory laterality (AI)	0.36 (0.40)	−0.06 (0.66)	−0.22 (0.66)	0.26 (0.58)	0.04 †

Data are shown as mean (SD), unless stated otherwise. Abbreviations: SD, standard deviation; UPCB, unilateral posterior crossbite. Analyzed by a Chi square test between control, left UPCB and right UPCB groups. † Kruskal–Wallis test. Significance values have been adjusted by the Bonferroni correction for multiple tests. Midline deviation refers only on the magnitude, not in the direction.

**Table 2 ijerph-17-05303-t002:** Reproducibility estimated by the intraclass correlation coefficients for the main variables regarding the mandibular position.

Reproducibility		Intraclass Correlation Coefficients (95%CI)			
		Intercuspal Position	Rest Position	Left Position	Right Position
Horizontal acromion alignment	Standing	0.71 (0.25;0.90)	0.72 (0.26;0.91)	0.76 (0.38;0.92)	0.82 (0.50;0.94)
	Seated	0.88 (0.44;0.97)	0.75 (0.05;0.94)	0.72 (0.26;0.91)	0.79 (0.43;0.93)
Horizontal ASIS alignment	Standing	0.61 (0.08;0.87)	0.53 (0.02;0.83)	0.41 (−0.10;0.77)	0.54 (0.04;0.84)
	Seated	-	-	-	-
CT angle	Standing	0.75 (0.34;0.92)	0.78 (0.40;0.93)	0.89 (0.58;0.97)	0.77 (0.40;0.93)
	Seated	0.78 (0.41;0.93)	0.78 (0.43;0.93)	0.68 (0.19;0.90)	0.73 (0.28;0.91)
Weight-foot distribution	Standing	0.64 (0.16;0.88)	0.61 (0.13;0.87)	0.68 (0.23;0.90)	0.58 (0.01;0.86)
	Seated	-	-	-	-

Abbreviations: ASIS, anterior-superior iliac spine; CI, confidence interval; CT, cervical-tragus.

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
