# Peer review of "Relationship between Unilateral Posterior Crossbite and Human Static Body Posture"

_ijerph, 2020, doi:10.3390/ijerph17155303_

Round 1

Reviewer 1 Report

No further comments from my end.

Reviewer 2 Report

The manuscript improves considerably, thank you for responding the suggestions, now I recommend this manuscript for publication

This manuscript is a resubmission of an earlier submission. The following is a list of the peer review reports and author responses from that submission.

Round 1

Reviewer 1 Report

Line 66-67, I think that is not necessary describing the null hypothesis.   Line 72, What was the inclusion and elimination criteria?   Line 78-79. I did not understand completely this paragraph. Did the authors include patients with same characteristics as is described at the exclusion criteria?   line 81. Why did the authors decide to include only one researcher to evaluate the measures? Would it be possible to use a second researcher to evaluate the measures, and make double blind evaluation, measured by Kappa? I think it would provide more strength to the design of the study.   Line 85, I did not understand the meaning of 10-cm VAS, could they explain this?   Line 138. I do not understand completely this paragraph, the authors described that they included a total of 36 adults and then 12 of them were including to test-retest, please explain it in a better manner.   overall comments.   The authors made a great effort to evaluate adequately each variants of the study. The discussion and conclusion are well documented and explain in detail the limitations of the study. I have some questions that  the authors need to resolve. Thus, I recommend it with minor changes.

Author Response

*Reviewer 1.

Thank you for these comments. A revised version has been submitted improving the manuscript according to the reviewer’s suggestions.

Comment 1:

Line 66-67, I think that is not necessary describing the null hypothesis.  

Response: We agree with the reviewer. The primary and secondary aims of the study were previously described, then the null hypothesis is not necessary to describe.

Text change: The sentence “The null hypotheses was that no differences in body posture exist between young adults with and without UPCB.” has been deleted.

Comment 2:

Line 72, What was the inclusion and elimination criteria?  

Response: Authors agree with the reviewer that in the original text we only specified the exclusion criteria. In the revised version, a new sentence has been added to clarify the inclusion criteria in line 71.

Text change: “The inclusion criteria were young adults with a minimum of 28 natural teeth, and for the unilateral crossbite group (UPCB), they had at least 1 posterior teeth in crossbite in only one side.”

Comment 3:

Line 78-79. I did not understand completely this paragraph. Did the authors include patients with same characteristics as is described at the exclusion criteria?  

Response: The interview described in this paragraph was performed in order to apply the inclusion/exclusion criteria, and to collect data from participants. Applicants to the study who met the exclusion criteria within the interview, were excluded.

Text change: “presence of temporomandibular or painful musculoskeletal disorders, auditory, visual, vestibular, and neurological disorders” has been deleted from this paragraph in line 79.

Comment 4:

line 81. Why did the authors decide to include only one researcher to evaluate the measures? Would it be possible to use a second researcher to evaluate the measures, and make double blind evaluation, measured by Kappa? I think it would provide more strength to the design of the study.  

Response: We thank the reviewer for this comment. However, laterality tests were extensively used with a high reproductibility (Rovira-Lastra et al.).

Comment 5:

Line 85, I did not understand the meaning of 10-cm VAS, could they explain this?  

Response: We agree with the reviewer. The term VAS has been explained in detail in the text.

Text change: (Visual Analogue Scale)

Comment 6:

Line 138. I do not understand completely this paragraph, the authors described that they included a total of 36 adults and then 12 of them were including to test-retest, please explain it in a better manner.  

Response: Twelve of the 36 participants were invited on the test-retest to assess the reliability. We have modified the Materials and Methods section in line 124.

Text change:  A test-retest assessment was performed in 12 participants based on their availability.

Comment 7:

Overall comments.   The authors made a great effort to evaluate adequately each variants of the study. The discussion and conclusion are well documented and explain in detail the limitations of the study. I have some questions that the authors need to resolve. Thus, I recommend it with minor changes.

Response: We appreciate the reviewer comment.

Reviewer 2 Report

The questions asked by the authors are interesting and are the subject of much debate. Please find some comments below:

1) With regard to the error of the method, the fact that only 2 of the 12 participants were from the UPCB group may be problematic. This should be mentioned by the authors.

2) Also, with regard to the error of the method, the authors chose to report this using intraclass correlation coefficients. I would urge the authors also to present random error using the Dahlberg formula (or another method as per the preference of the authors), as this is able to quantify the error. This is especially important as differences found in the present study are small and perhaps not clinically relevant. If these differences are inferior to the random error then this implies that one cannot attribute any importance to these differences.

3) The sample size was calculated to be 18 for the control group and 18 for the UPCB group (regardless of side) and thus any comparisons with only right-sided or left-sided UPCB are underpowered. This should be stressed by the authors.

4) This is my most important concern and has to do with interpretation of the results. If correction for multiple testing is carried out, then it turns out that none of the differences are statistically significant. This should be done. This will, in consequence, change the overall findings, discussion, and conclusions since the conclusion will be that no significant differences were found between controls and UPCB.

Author Response

*Reviewer 2.

Thank you for these comments. A revised version has been submitted improving the manuscript according to the reviewer’s suggestions.

1) With regard to the error of the method, the fact that only 2 of the 12 participants were from the UPCB group may be problematic. This should be mentioned by the authors.

Response: Authors appreciate the reviewer’s comment. However, when we designed the study we did not expect that the error of the method was different between the control group and the UPCB group. In fact, when we analyzed the test-retest results we did not observe any difference between groups.

Text change: None.

2) Also, with regard to the error of the method, the authors chose to report this using intraclass correlation coefficients. I would urge the authors also to present random error using the Dahlberg formula (or another method as per the preference of the authors), as this is able to quantify the error. This is especially important as differences found in the present study are small and perhaps not clinically relevant. If these differences are inferior to the random error then this implies that one cannot attribute any importance to these differences.

Response: Thank you for your comment. The Dahlberg’s formula was calculated to complement the ICC values.

Text change: In the Materials and Methods section, line 126, the sentence “To calculate the measurement error, the Dahlberg’s formula was applied [29].” and a new cite has been added.

In the Results section, line 142, the sentence: “The Dahlberg error in measuring the horizontal alignment of the acromions ranged, depending on the mandibular position, from 1.2° to 1.6° by standing posture and from 0.6° to 1.1° by seated posture, and from 1.0° to 1.3° measuring the horizontal alignment of the ASIS.” has been added.

3) The sample size was calculated to be 18 for the control group and 18 for the UPCB group (regardless of side) and thus any comparisons with only right-sided or left-sided UPCB are underpowered. This should be stressed by the authors.

Response: We agree with the reviewer. The sample size was calculated with regard to the primary aim. Then, the findings about comparisons between right-side or left-side UPCB groups should be interpreted with caution.

Text change: In the Discussion section, line 204, the sentence “Although the sample size was sufficient to find body posture differences in subjects with and without UPCB, the small number of participants in the right-side or left-side UPCB group might not have been large enough to find significant comparisons and the findings should be interpreted with caution.” has been added.

4) This is my most important concern and has to do with interpretation of the results. If correction for multiple testing is carried out, then it turns out that none of the differences are statistically significant. This should be done. This will, in consequence, change the overall findings, discussion, and conclusions since the conclusion will be that no significant differences were found between controls and UPCB.

Response: We thank the reviewer for this comment. We applied correction for multiple testing by least significant difference (LSD) to estimate the marginal means for each general linear model with repeated measures. We also applied the Bonferroni correction while using the Kruskal-Wallis test to analyze differences in horizontal alignments, demographic characteristics and functional lateralities between control, left UPCB and right UPCB groups.

Text change: None.

Round 2

Reviewer 2 Report

The authors have adequately responded to the first three criticisms. I do however insist on my main concern with regard to the interpretation of the data. I will reinsert my initial comment here: "This is my most important concern and has to do with interpretation of the results. If correction for multiple testing is carried out, then it turns out that none of the differences are statistically significant. This should be done. This will, in consequence, change the overall findings, discussion, and conclusions since the conclusion will be that no significant differences were found between controls and UPCB". According to the authors response, they claim to have applied correction for multiple testing by least significant difference (LSD). Unlike tests such as Bonferroni, Tukey, Dunnett and Holm methods, LSD does not correct for multiple comparisons. If one chooses to use the LSD test, one will need to account for multiple comparisons when interpreting the data, since the computations themselves do not correct for multiple comparisons. The authors then go on to say that they applied Bonferroni correction, however when looking at the interpretation of the p-values this does not seem to be the case. For example, they consider that a p-value of 0.035 is statistically significant. However, even if only 2 tests were carried out, following Bonferroni correction, a p-value of 0.025 would be considered statistically significant. Obviously more than 2 tests were carried out here and thus the real correction would produce an even smaller p-value that should be considered statistically significant. Lastly, the authors should look at differences between groups, and if a difference between groups is inferior to or equal to the error (that they calculated using Dahlberg's formula) then this surely would also influence the interpretation of their data. I suggest that authors apply Bonferroni correction, look at their data in light of the error of the method, and reinterpret and rediscuss their findings.